# Phylogeny and Genetic Divergence among Sorghum Mosaic Virus Isolates Infecting Sugarcane

**DOI:** 10.3390/plants12213759

**Published:** 2023-11-02

**Authors:** Hui-Mei Xu, Er-Qi He, Zu-Li Yang, Zheng-Wang Bi, Wen-Qing Bao, Sheng-Ren Sun, Jia-Ju Lu, San-Ji Gao

**Affiliations:** 1National Engineering Research Center for Sugarcane, Fujian Agriculture and Forestry University, Fuzhou 350002, China; xhmxhm946946@163.com (H.-M.X.); sdwlbzw@163.com (Z.-W.B.); wenqingbao0706@163.com (W.-Q.B.); 2Guizhou Institute of Subtropical Crops, Guizhou Academy of Agricultural Sciences, Xingyi 562400, China; heerqi003@163.com; 3Laibin Academy of Agricultural Sciences, Laibin 546100, China; yang_zuli@163.com; 4Institute of Nanfan & Seed Industry, Guangdong Academy of Sciences, Guangzhou 510316, China; ssr03@163.com

**Keywords:** genetic diversity, molecular evolution, mosaic disease, population structure, sorghum mosaic virus, sugarcane

## Abstract

Sorghum mosaic virus (SrMV, the genus *Potyvirus* of the family *Potyviridae*) is a causal agent of common mosaic in sugarcane and poses a threat to the global sugar industry. In this study, a total of 901 sugarcane leaf samples with mosaic symptom were collected from eight provinces in China and were detected via RT-PCR using a primer pair specific to the SrMV coat protein (*CP*). These leaf samples included 839 samples from modern cultivars (*Saccharum* spp. hybrids) and 62 samples from chewing cane (*S. officinarum*). Among these, 632 out of 901 (70.1%) samples were tested positive for SrMV. The incidences of SrMV infection were 72.3% and 40.3% in modern cultivars and chewing cane, respectively. Phylogenetic analysis showed that all tested SrMV isolates were clustered into three clades consisting of six phylogenetic groups based on 306 *CP* sequences (this study = 265 and GenBank database = 41). A total of 10 SrMV isolates from South America (the United States and Argentina) along with 106 isolates from China were clustered in group D, while the remaining 190 SrMV isolates from Asia (China and Vietnam) were dispersed in five groups. The SrMV isolates in group F were limited to Yunnan province in China, and those in group A were spread over eight provinces. A significant genetic heterogeneity was elucidated in the nucleotide sequence identities of all SrMV CPs, ranging from 69.0% to 100%. A potential recombination event was postulated among SrMV isolates based on *CP* sequences. All tested SrMV *CP*s underwent dominant negative selection. Geographical isolation (South America vs. Asia) and host types (modern cultivars vs. chewing cane) are important factors promoting the genetic differentiation of SrMV populations. Overall, this study contributes to the global understanding of the genetic evolution of SrMV and provides a valuable resource for the epidemiology and management of the mosaic in sugarcane.

## 1. Introduction

The *Potyviridae* is the largest family of known RNA viruses, having significant impact on agriculture and ecology [1,2,3]. There are currently 12 genera and 246 species in this virus family (https://ictv.global/taxonomy (accessed on 28 October 2023). The largest genus of plant viruses, the *Potyvirus*, belongs to the *Potyviridae* family and causes significant losses in a variety of crops around the world, such as maize, potatoes, sorghum, soybeans, sugarcane, and so on [4,5,6,7]. Mosaic is an important viral disease of sugarcane around the world, caused by a single or mixed infection of sugarcane mosaic virus (SCMV), sorghum mosaic virus (SrMV), sugarcane streak mosaic virus (SCSMV), and maize yellow mosaic virus (MaYMV) [8,9]. In addition to sugarcane, these viruses have a wide range of hosts such as maize, sorghum, and other grasses [8]. SCMV and SrMV (genus *Potyvirus*, family *Potyviridae*) are distributed globally, while SCSMV (genus *Poacevirus*, family *Potyviridae*) appears in Asia and Côte d’Ivoire and MaYMV (genus *Ampelovirus*, family *Closteroviridae*) occurs in Africa, Asia, as well as South America [9,10,11,12].

The potyviral genome encodes a long polyprotein that is processed by proteinases, giving rise to at least 10 mature proteins: P1 (protein 1 protease), HC-Pro (helper component protease), P3 (protein 3), PIPO (pretty interesting Potyviridae ORF), 6K1 (6 kDa peptide 1), CI (cylindrical inclusion), 6K2 (6 kDa peptide 2), NIa-Pro (nuclear inclusion a-protease), NIb (nuclear inclusion b, RNA-directed RNA polymerase), CP (capsid protein), as well as VPg (virus protein genome-linked) [5,13,14]. The potyviral CP participates in various biological functions such as coating and protection of the RNA genome, aphid transmission, as well as cell-to-cell and long-distance movement [4,15]. Meanwhile, this viral protein may also be involved in the regulation of CP stability and functional diversity during the viral life cycle through various post-translational modifications [4]. However, these biological functions in SrMV CP have not yet been identified. The CP-coding region of these potyvirus-encoded proteins is preferentially a targeted region used for viral genetic diversity and phylogeny analysis as well as disease diagnosis using molecular and serological approaches [5,8,16].

The genetic diversity of SrMV isolates has been explored using sequencing technology and virus taxonomy. In the 1990s, three strains of SrMV (H, I, and M) were identified, and these were distinguished from SCMV, johnsongrass mosaic virus (JGMV), and maize dwarf mosaic virus (MDMV) [17,18,19]. Perera et al. reported that the CP nucleotide identities ranged from 97.4% to 99.9% among SrMV isolates from Argentina [20]. In China, based on *CP* sequence analysis, Xu et al. (2008) revealed that obvious genetic diversity (76–100%) occurred among 18 SrMV isolates, while Luo et al. (2016) found that the nucleotide identities were 74.3–94.1% (nucleotide) and 84.7%–98.1% (amino acid) among 188 SrMV isolates worldwide [21]. Meanwhile, Zhou et al. (2014) demonstrated that SrMV-GX together with SrMV-XoS, SrMV-YH, and SrMV-H were grouped in the same evolutionary cluster based on the genomic sequence analysis, and they shared sequence identities of 80.9–95.4% and 90.4–98.4% at nucleotide and amino acid levels [22], respectively. Several phylogenetic groups of SrMV isolates were proposed based on viral CP and genome sequences. For example, two phylogenetic groups of SrMV were clustered based on host origins, i.e., modern cultivars (*Saccharum* hybrids spp.) vs. noble cane (*S. officinarum*) [21,23]. Three and six phylogenetic groups of SrMV were proposed by Zhang et al. (2015) [24] and Luo et al. (2016) [25], respectively, in China.

Evolutionary driving forces, population structure, and differentiation among SrMV isolates have been investigated. For instance, insertion/deletion mutations, negative selection, and frequent gene flow were proposed to contribute to the genetic divergence and population structure of SrMV isolates [25]. No obvious recombination event was found in the *CP* gene region of all tested SrMV isolates [24,25]. High rates of mutation and recombination between potyvirus strains result in the creation of new viral strains. These novel isolates show a high degree of pathogenicity in a variety of host species and cultivars, which is posing a challenge to global crop production [4,26,27]. Importantly, it is critical to distinguish between SrMV strains when breeding resistant sugarcane genotypes. However, these research aspects of SrMV remain unclear. In China, SrMV is one of main viruses infecting sugarcane, particularly modern commercial cultivars, followed by SCSMV [8,25,28]. Therefore, in this study we extensively analyze the occurrence, distribution, genetic variation, and population differentiation of SrMV infecting sugarcane based on viral *CP* fragments. These findings offer insights into the virus’s prevalence in China’s sugarcane-growing regions and crucial recommendations for the management and prevention of mosaic disease.

## 2. Results

### 2.1. Detection of SrMV Using RT-PCR

The SrMV was detected using RT-PCR in 632 of 901 (70.1%) leaf samples. The incidences of SrMV-positive were 72.3% and 40.3% in modern cultivars and chewing cane, respectively. Subsequently, 265 representative *CP* fragments (approximately 850 bp) were selected for a further sequence analysis.

### 2.2. Phylogenetic Relationship among SrMV Isolates

A phylogenetic analysis showed that all the 306 SrMV isolates (this study = 265 and GeneBank library = 41) were clustered into three clades (I, II, and III), including six different groups (A–F) with 4–120 isolates. Clades I and II consisted of two (A and B) and three (C–E) groups, respectively. Clades III included a unique group F. Moreover, 39.2% and 37.9% of SrMV isolates were assigned to groups A and D, respectively. Apart from 106 SrMV isolates from China, 10 isolates from the United States and Argentina were clustered into group D. The remaining SrMV isolates from Asia (China and Vietnam) were clustered in six groups (Figure 1). Notably, the 18 SrMV isolates from chewing cane were distributed in four groups (SrMV-A, -D, -E, and -F). The frequency of SrMV phylogroups over eight Chinese sugarcane-planting provinces is shown in Figure 2. The SrMV isolates from groups A and D were observed in eight provinces, while the SrMV isolates from group B were found in seven provinces except Guangdong (GD). Additionally, the SrMV isolates from group E were found in six provinces except Sichuan (SC) and Yunnan (YN) provinces. Group C was present in four provinces including Fujian (FJ), Guangxi (GX), Hainan (HN), and Sichuan, but group F only occurred in Yunnan province.

### 2.3. Sequence Identities between SrMV Populations

The sequence identities of 265 SrMV isolates obtained in this study ranged from 70.3 to 100% (nucleotide) and from 73.8 to 100% (amino acid). In each phylogenetic group, the minimum sequence identities of 73.2% (nucleotide) and 80.5% (amino acid) were observed between SCJK003 (MZ419743) and other isolates in group D (Table 1). Among six phylogenetic groups, nucleotide sequence identities ranged between 69.0% (between groups D and E) and 97.5% (between groups C and D), while amino acid sequence identities were 72.8% (between groups A and D) and 100% (between groups C and D). Notably, obvious divergence was exhibited between clade I and the other two clades (II and III), as evidenced by lower nucleotide sequence identities (<85%) among SrMV isolates, except those between the SCJK003 isolate (group D) and 111 SrMV isolates in group A. In addition, the nucleotide and amino acid sequence identities between geographical groups were between 71.3–100% and 74.9–100%, respectively (Appendix A). Meanwhile, nucleotide and amino acid identities between host origin groups were shared by 71.3–100% and 74.9–100%, respectively (Appendix A).

To further investigate the variation among SrMV *CP* sequences, 12 representative CP amino acid sequences (two sequences in each phylogroup) were aligned. At least four Insertion/deletion (InDel) at the N-terminal and 26 mutation sites were exhibited in SrMV CP sequences (Appendix A). It is noteworthy that no deletion, but a unique site mutation, was present among these CP amino acid sequences in the SrMV-F group as compared to other groups.

### 2.4. Genetic Recombination Events among SrMV Isolates

Genetic recombination events were identified using RDP4 based on 306 *CP* sequences of SrMV. A significant recombination event was found as supported by four algorithms (*p* < 0.05). The potential recombinant isolate was FJSX017 (MZ419585) from Fujian province, China. The recombinant was derived from the major parent SCH (U07219) from the United States and a minor unknown parent (Table 2).

### 2.5. Neutrality Test and Selection Pressure on SrMV Populations

Nucleotide diversity (π) showed that SrMV CP sequences in the Asian population had a higher genetic variation (π = 0.12160), while the sequences in the American population had a lower genetic variation (π = 0.02200). However, the π values of the SrMV CP sequences in modern cultivars and chewing cane were 0.12062 and 0.10265, respectively, indicating a higher genetic variation of SrMV CP in both host origins. A neutrality test showed that Tajima’s D values for four SrMV populations were all negative, suggesting that the SrMV population exhibited a trend of expansion. Conversely, Tajima’s D values for neutrality tests were not statistically significant (*p* > 0.10) in all cases. Meanwhile, the ratios of dN/dS ranged from 0.070 to 0.078 (less than 1) among four populations, suggesting that the SrMV CP gene was under a negative selection (Table 3).

### 2.6. Genetic Differentiation and Gene Flow between SrMV Populations

Geographic (Asia vs. America) and host (modern cultivars vs. chewing cane) origins showed considerable genetic differentiation as detected by three permutation-based statistical tests (Ks*, Z*, and Snn) that reached significant levels (*p* < 0.05). The Fst values were >0.33, and the Nm values were <1.0 between geographical groups (Asia vs. America), suggesting that the gene flow between two populations was not frequent. Conversely, the Fst values were <0.33, and the Nm values were >1.0 between host origins (modern cultivars vs. chewing cane), indicating that the gene flow between two populations was frequent (Table 4).

## 3. Discussion

The crop productivity is affected by a wide range of adverse environmental factors, including biotic and abiotic stress [29]. Mosaic can cause losses ranging from 17% to 50% in susceptible varieties [8]. A survey of the occurrence and distribution of causal agents is an important step for prevention and control for this disease. However, SrMV is often mixed with other viruses causing mosaic diseases, and, therefore, distinguishing the species or strain of viruses causing mosaic disease is nearly impossible through a visual observation [8,25,30,31]. In this study, the RT-PCR technology was employed to accurately identify SrMV. A higher SrMV detection rate was found in modern cultivars than chewing cane. A lower SrMV detection rate existed in chewing cane because of the lower vulnerability of the host to SrMV pathogenesis [21,32]. In addition to cultivar resistance, vector populations and their vagility being subjected to ecosystem simplification also affects virus infection rates [33]. However, this difference in interaction between the virus and sugarcane host need to be further explored. In addition to SrMV, SCSMV is another main causal agent of mosaic in sugarcane modern cultivars in China [8,21,22]. 

According to our findings, the SrMV isolates from China and South America were grouped together in the SrMV-D group, while the SrMV isolates from Asia were distributed throughout the six phylogroups. Compared to a previous study by Luo et al. (2016), a large number of sugarcane samples was used in this study, but no new phylogroup was proposed [25]. Nonetheless, more phylogroups were discovered in some specific provincial regions in China. For example, Luo et al. (2016) proposed that only one phylogroup (SrMV-G1) occurred in Guizhou province, while the results of the current study indicate that four phylogroups (SrMV-A, -B, -D, and -E) were in this region [25]. The possible reason is that more leaf samples with mosaic were analyzed in this study. The low sequence identities among the SrMV isolates were indicative of high genetic divergence. The viral species demarcation in *Potyviridae* family is typically based on the sequence identity of the CP-coding region with a threshold of <76% (nucleotide) and <82% (amid acid) [34]. Therefore, even if these isolates are in line with the threshold of viral species in this family, more research is required to determine whether SrMV isolates from Yunnan Province clustered in phylogroup F belong to a unique quasispecies or species. New viral species will be considered following investigations based on full genome sequences (genomic feature and phylogeny), together with additional data about biological characteristics such as host range and vector [5]. 

A high rate of viral genome mutation aids in the creation of novel strains, including resistance-breaking isolates [4]. Our data showed that there are at least four InDels in the N-terminal of SrMV CPs and numerous site mutations across the viral CP sequence. Notably, an obvious feature of CP amino acid sequences in the SrMV-F group is no deletion, but a unique mutation site is present compared to other groups. It is unclear whether these different SrMV isolates in different phylogroups are associated with the variation of viral pathogenicity. Additionally, recombination is a major driving force in the evolution of potyviruses [35,36], but this evolutionary force seems to be an uncommon mechanism of speciation [35]. Our data showed that there was a potential recombination event in all tested SrMV *CP* sequences. However, no recombination was found in previous studies by Zhang et al. (2015) [24] and Luo et al. (2016) [25]. Natural selection is another important evolutionary mechanism and driving force for viral population variation, and purification selection accelerates the elimination of harmful mutations in genes as well as the formation of a stable population genetic structure [37]. Notably, all potyvirus genomes undergo a negative selection, with certain genes such as *HC-Pro*, *CP*, *Nia*, and *NIb* being more strongly selected than others [35]. In this study, the tested SrMV *CP* was subjected to negative selection.

The genetic makeup of viral populations is significantly influenced by geographic isolation [37]. However, modern travel and trade have grown to be significant factors in the transmission of viruses and the swapping of their hosts [35]. Sugarcane is a vegetative propagated crop and frequent exchange of germplasm resources or plant settings between Asian countries, which likely resulted in the absence of obvious population divergence of SrMV within the Asian population. Similarly, Wang et al. (2017) also demonstrated that there was no obvious geographic difference among SrMV isolates [38]. But, to some extent, SrMV populations in China were linked to their geographical origins [24]. Here, our data showed that geographic isolation plays a significant role in the divergence of SrMV isolates between Asia and South America. The host type is another crucial factor leading to the genetic differentiation of plant viruses [33,37]. Based on the phylogenetic analysis of 18 Chinese SrMV isolates, they were divided into two virus populations associated with host types (moder cultivar and chewing cane) [21]. Our data revealed that the phylogenetic grouping of SrMV isolates was not related to two host sources. On the other hand, these SrMV isolates were strongly differentiating the populations of chewing cane and modern cultivars, according to genetic differentiation analysis. A large-scale study of SrMV samples, host sources, and sugarcane-planting regions should be carried out to further explore SrMV population differentiation. Overall, various driving forces contribute to form different SrMV populations or quasispecies.

## 4. Materials and Methods

### 4.1. Collection and Distribution of Leaf Samples

A total of 901 leaf samples with mosaic were collected from eight sugarcane-planting provinces from 2017 to 2020, including 839 samples from modern cultivars (*Saccharum* spp. hybrids, Sh) and 62 samples from chewing cane (*S. officinarum*, So). Distribution of leaf samples in different provinces: Guangdong (Sh = 45), Guangxi (Sh = 173), Guizhou (Sh = 165 and So = 5), Fujian (Sh = 90 and So = 9), Hainan (Sh = 51), Sichuan (Sh = 181), Yunnan (Sh = 81), and Zhejiang (Sh = 53 and So = 48). All leaf samples were scrubbed and disinfected with 75% alcohol and stored at −80 °C for further molecular detection.

### 4.2. RT-PCR Detection

Total RNA was extracted from leaf samples using the TRIzol^®^ Reagent (Invitrogen, Carlsbad, CA, USA). After the quality and quantity of total RNA were checked, these RNA samples were used for molecular detection using RT-PCR [39]. The HiScript II 1st Strand cDNA Synthesis Kit (Novozan, Nanjing, China) was used to synthesize cDNA from each RNA sample (1.0 µg) with the reverse transcription primer Oligo (dT) _23_VN. The set of SrMV-specific primers SrMV-F (5′-ACAGCAGAWGCAACRGCACAAGC-3′) and SrMV-R (5′-CTCWCCGACATTCCCATCCAAGCC-3′) was used for PCR amplification [39]. The PCR amplification in a 25 µL volume included 1 µL cDNA, 12.5 µL Premix Taq (Ex Taq Version 2.0 plus dye) (TaKaRa, Dalian, China), and 1 µL of each primer (10 µmol/L). The PCR was performed in the following conditions: 94 °C for 5 min; followed by 35 cycles at 94 °C for 30 s, 52 °C for 30 s, and 72 °C for 1 min; a final extension at 72 °C for 10 min. The PCR products were analyzed via a gel electrophoresis on 1.0% agarose gels.

### 4.3. Cloning and Sequencing of RT-PCR Fragments

The target fragments from partial SrMV-positive PCR products were eluted using a Gel Extraction Kit (OMEGA Bio-Tek, Norcross, GA, USA). The purified PCR fragments were ligated into the pMD19-T vector (TaKaRa) and then transformed into *Escherichia coli* DH5α competent cells. Three positive colonies from each leaf sample were sent to Sangon Biotech Co., Ltd. (Shanghai, China) for sequencing. The inserted fragments were sequenced bidirectionally using the M13 universal primers.

### 4.4. Sequence Alignment and Phylogenetic Analysis

A total of 306 CP sequences (this study = 265 and GenBank database = 41) trimming the primer pair sequences were used for sequence alignment and phylogenetic analysis, including 288 sequences from modern cultivars and 28 sequences from chewing cane (Appendix A). Sequence alignment was carried out using the ClustalW algorithm implemented in MEGA 10.1.8 software [40] The Neighbor-joining (NJ) method was used to construct the phylogenetic tree, and the robustness of the nodes of the phylogenetic tree was assessed from 1000 bootstrap replicates. A sequence of SCMV isolate SCMV-HZ (NC_003398) was used as outgroup. Sequence identity analysis was conducted using BioEdit 7.1.9 software [41].

### 4.5. Genetic Recombination Analysis

Sequence recombination analysis was performed using seven different recombination algorithms (RDP, GENECONV, Chimaera, MaxChi, Bootscane, SISCAN, and 3Seq) implemented in RDP4 (Recombination Detection Program version 4) software [42]. Only recombination events that were detected by more than four algorithms (*p* < 0.05) were considered significant.

### 4.6. Evaluation of Population Genetic Parameters

All population genetic parameters of SrMV *CP* sequences based on different geographical origins (South America vs. Asia) and sugarcane hosts (modern cultivar vs. chewing cane) were calculated using DnaSP version 5.10.01 software [43]. Genetic parameters included nucleotide diversity (π) [44] and Tajima’s D [45]. Three statistical test values (Ks*, Z*, and Snn) were used to evaluate the genetic differentiation between SrMV populations. |Fst| > 0.33 or Nm < 1 indicates that gene flow between populations is not frequent, while |Fst| < 0.33 or Nm > 1 suggests frequent gene flow. The selection pressure on SrMV CP in each population was evaluated by calculating the ratio of nonsynonymous (dN) and synonymous (dS) substitutions in nucleotide sequences. Positive, neutral, and negative selections were indicated by dN/dS ratios >1, =1, and <1.

## 5. Conclusions

In this study, the molecular divergence and population structure of SrMV isolates infecting sugarcane (modern cultivars and ancient chewing cane) were investigated based on the *CP* fragment sequences. A high incidence (70.1%) of the samples was tested positive for SrMV through an RT-PCR assay. Based on 306 SrMV *CP* sequences, three clades including six phylogenetic groups were proposed. High genetic diversity was present among all tested SrMV isolates based on *CP* sequence identities ranging from 69.0% to 100%. The SrMV-A and -D groups dispersed in eight sugarcane planting regions/provinces, but SrMV-F only occurred in Yunnan province, China. Our data suggested that numerous evolutionary driving forces such as nucleotide mutant, gene recombination, and purifying selection as well as geographical and host isolation contributed to form different SrMV populations around the world. These findings enrich the information of the genetic diversity of this virus. However, the molecular divergence and genetic population of SrMV at the complete genome level remain unclear. In addition, the molecular mechanism of the interaction between this virus and host sugarcane is largely unknown. Therefore, these research aspects need to be further explored.

## Figures and Tables

**Figure 1 plants-12-03759-f001:**
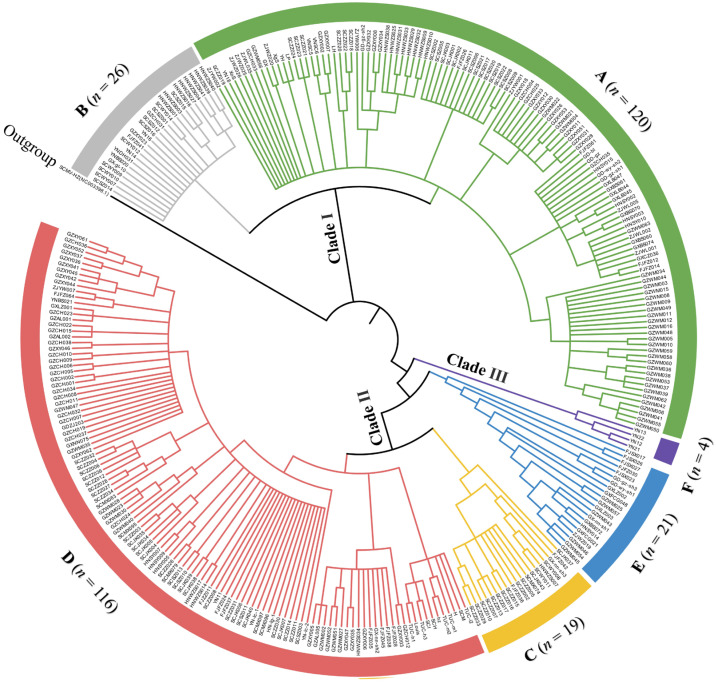
Phylogenetic tree based on nucleotide sequences of coat protein (*CP*) from 306 sorghum mosaic virus (SrMV) isolates. All tested SrMV CP sequences included 265 isolates from this study plus 41 isolates from the GenBank database. A sequence of sugarcane mosaic virus (SCMV) isolate SCMV-HZ (GenBank accession no. NC_003398) was used as outgroup. The number (*n*) of isolates in each phylogroup is in parentheses.

**Figure 2 plants-12-03759-f002:**
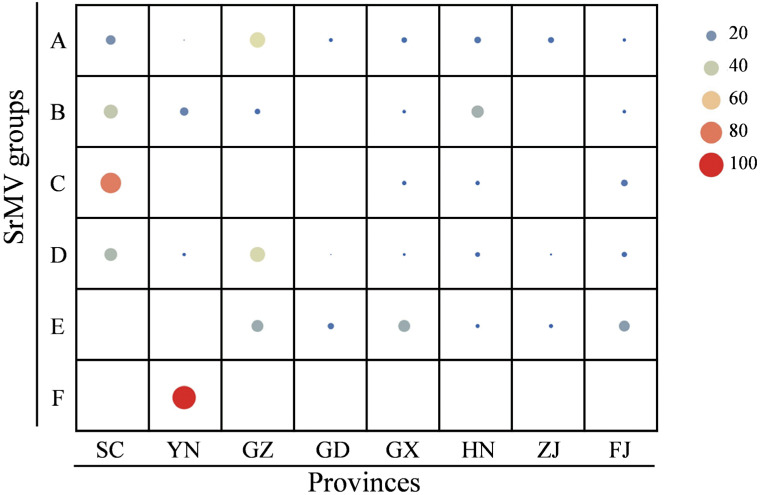
Distribution and frequency (%) of the SrMV phylogroups (A–F) over eight sugarcane planting provinces in China. FJ = Fujian (*n* = 99), GD = Guangdong (*n* = 45), GX = Guangxi (*n* = 173), GZ = Guizhou (*n* = 170), HN = Hainan (*n* = 51), SC = Sichuan (*n* = 181), YN = Yunnan (*n* = 81), and ZJ = Zhejiang (*n* = 101). The size of the circles represents the SrMV phylogroup frequency (%), wherein larger circles correspond to a higher frequency.

**Table 1 plants-12-03759-t001:** Percentage identities (%) of nucleotide (low-left) and amino acid (up-right) sequences of SrMV coat protein within and between phylogenetic groups ^a^.

Group	A (*n* = 120)	B (*n* = 26)	C (*n* = 19)	D (*n* = 116)	E (*n* = 21)	F (*n* = 4)
A	85.3–100 (81.1–100)	77.8–97.7	73.2–89.2	72.8–90.0	73.8–91.8	77.5–90.3
B	79.3–94.0	89.2–100 (88.5–99.6)	76.0–89.2	76.4–90.0	77.1–92.2	81.4–91.0
C	70.7–80.7	69.1–81.3	87.8–100 (86.1–99.5)	79.2–100	81.4–97.5	88.5–95.7
D	70.1–90.7	69.3–84.6	71.7–97.5	80.5–100 (73.2–100)	75.0–98.5	82.5–96.4
E	72.5–81.8	69.4–82.5	78.9–92.4	69.0–92.6	86.7–100 (82.4–100)	84.6–96.0
F	71.6–80.3	72.2–81.2	80.3–87.3	75.2–87.4	78.0–87.3	97.1–99.6 (95.6–99.5)

^a^ Nucleotide sequence identities (%) of SrMV CP within phylogenetic groups are shown in parentheses.

**Table 2 plants-12-03759-t002:** Recombination signals detection among 306 SrMV isolates based on the *CP* genes.

Recombinant	Potential Parents		Detection Method ^a^						
	Main Parent	Minor Parent	R	G	B	M	C	S	T
FJSX017 (MZ419585)	SCH (U07219)	Unknown	-	-	-	+	+	+	+

^a^ Seven algorithms include RDP (R), GENECONV (G), Booscan (B), Maximum Chisquare (M), Chimaera (C), Sister Scan (S), and 3Seq (T); +, significant (10^−6^ < *p* ≤ 0.05); -, non-significant (*p* > 0.05).

**Table 3 plants-12-03759-t003:** Genetic variation and population genetic parameters between different populations based on SrMV *CP* sequences.

Population	π	Tajima’s D ^a^	dN/dS ^b^
Total (*n* = 306)	0.12186	−0.28177 (ns)	0.077
Asia (*n* = 296)	0.12160	−0.25049 (ns)	0.078
South America (*n* = 10)	0.02200	−1.25048 (ns)	0.077
Modern cultivar (*n* = 288)	0.12062	−0.28584 (ns)	0.078
Chewing cane (*n* = 18)	0.10265	−0.61461 (ns)	0.070

^a^ ns, non-significant. ^b^ dN/dS, the ratio of nonsynonymous (dN) and synonymous (dS) substitution.

**Table 4 plants-12-03759-t004:** Tests of genetic differentiation and gene flow among SrMV groups based on geographical origins and host types ^a^.

Comparison	Ks* (*p*-Value)	Z* (*p*-Value)	Snn (*p*-Value)	Fst	Nm
Asia (*n* = 296) vs. South America (*n* = 10)	4.14107 (0.0000 ***)	9.68253 (0.0000 ***)	0.99183 (0.0000 ***)	0.44293	0.63
Modern cultivar (*n* = 288) vs. chewing cane (*n* = 18)	4.15947 (0.0000 ***)	9.71312 (0.0000 ***)	0.92157 (0.0300 *)	0.15628	2.70

^a^ Asterisks: *, 0.01 < *p* < 0.05; ***, *p* < 0.001.

## Data Availability

All data supporting the findings of this study are available within the paper and its Appendix A.

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
