# Peer review of "Phylogeny and Genetic Divergence among Sorghum Mosaic Virus Isolates Infecting Sugarcane"

_plants, 2023, doi:10.3390/plants12213759_

Round 1

Reviewer 1 Report

Comments and Suggestions for Authors

Thank you for considering me to revise the manuscript titled “Phylogeny and genetic divergence among sorghum mosaic virus isolate infecting sugarcane”. The manuscript provides a fairly robust dataset on exploring the genetic divergence and phylogeny among mosaic virus isolates collected from infected sugarcane plants in different provinces of China during four years. The obtained results are beneficial for genetic evolution of SrMV as well as for breeding resistant genotypes to mosaic virus in sugarcane. In general, the manuscript is well-structured and could be accepted after minor revision.

Suggestions

The manuscript needs major English editing, several grammatical errors, and long sentences should be revised throughout the manuscript.

The title could be improved to be “Genetic Divergence and Phylogeny among Sorghum Mosaic Virus Isolates Collected from Infected Sugarcane Plans in Different  Provinces of China”

More details on the applied methodology are needed in the abstract

Line 25: “Yunnna” should be replaced by “Yunnan”

The introduction needs to be improved, the citations need to be updated (Some from a long time ago as 1992, 1997, and 2008). The importance of exploring genetic divergence and phylogeny among sorghum mosaic virus isolates in sugarcane should be highlighted in the introduction. Also, the importance of distinguishing the virus strains in breeding resistant sugarcane genotypes. Moreover, the knowledge gap, rationale, and hypothesis should be clarified.

The results section should be improved and more efforts are needed from the authors to improve the obtained results. The number after decimals in Table 4 as well as Table 3 could be shortened to three numbers.

  The discussion is short, needs to be improved and the obtained results should be discussed better. More literature should be explored with the obtained results.

The section of material and methods is well-written 

Line 341: (Saccharum officinarum) should be in italics and all scientific names throughout the manuscript

Line 397: Genomics and Applied Biology should be abbreviated 

Comments on the Quality of English Language

Moderate editing of English language required

Author Response

Comment 1: Thank you for considering me to revise the manuscript titled “Phylogeny and genetic divergence among sorghum mosaic virus isolate infecting sugarcane”. The manuscript provides a fairly robust dataset on exploring the genetic divergence and phylogeny among mosaic virus isolates collected from infected sugarcane plants in different provinces of China during four years. The obtained results are beneficial for genetic evolution of SrMV as well as for breeding resistant genotypes to mosaic virus in sugarcane. In general, the manuscript is well-structured and could be accepted after minor revision. Reply: Thank you for your positive comments. Comment 2: The manuscript needs major English editing, several grammatical errors, and long sentences should be revised throughout the manuscript. Reply: We have seriously improved the English in the whole manuscript under the English-speaking expert. Comment 3: The title could be improved to be “Genetic Divergence and Phylogeny among Sorghum Mosaic Virus Isolates Collected from Infected Sugarcane Plans in Different Provinces of China”. Reply: Sorry, we prefer to keep the original title because SrMV CP sequences tested in this study including China and other countries such as the United States, Argentina, and Vietnam. Comment 4: More details on the applied methodology are needed in the abstract. Reply: We have added more methodology in the abstract. Comment 5: Line 25: “Yunnna” should be replaced by “Yunnan” Reply: Revised as you required. Comment 6: The introduction needs to be improved, the citations need to be updated (Some from a long time ago as 1992, 1997, and 2008). The importance of exploring genetic divergence and phylogeny among sorghum mosaic virus isolates in sugarcane should be highlighted in the introduction. Also, the importance of distinguishing the virus strains in breeding resistant sugarcane genotypes. Moreover, the knowledge gap, rationale, and hypothesis should be clarified. Reply: Revised as you required. For example, overview of Potyviruses and its impacts on agriculture and ecology, the function of potyviral CP, High mutation rates and recombination between different strains of potyviruses, and so on. Comment 7: The results section should be improved and more efforts are needed from the authors to improve the obtained results. The number after decimals in Table 4 as well as Table 3 could be shortened to three numbers. Reply: We added the content about insertion/deletion (InDel) and site mutation among SrMV CP sequences. We think it is fine the number after decimals in Table 4 and Table 3. Comment 8: The discussion is short, needs to be improved and the obtained results should be discussed better. More literature should be explored with the obtained results. Reply: We have improved this section. We discussed more the roles of high mutation rate of viral genome in potyviruses and various driving forces contributing to form different SrMV populations. Comment 9: The section of material and methods is well-written. Reply: Thank you for your positive comments. Comment 10: Line 341: (Saccharum officinarum) should be in italics and all scientific names throughout the manuscript. Reply: Revised as you required. Comment 11: Line 397: Genomics and Applied Biology should be abbreviated. Reply: Revised as “Genom. Appl. Biol.”.

Reviewer 2 Report

Comments and Suggestions for Authors

Comments for the Author:

In the manuscript of “Phylogeny and Genetic Divergence among Sorghum Mosaic Virus Isolates Infecting Sugarcane ”, Which enriches the information of genetic evolution of SrMV world-wide and provides a reference for epidemiology and control of mosaic in sugarcane. These findings enrich the information of genetic diversity of this virus. in my opinion, just a few changes are required.

1.      In lines 77-78, The author claims that SrMV is one of main viruses infecting sugarcane particularly modern commercial cultivars, followed by SCSMV in China. Are all the 901 materials used by the author modern commercial cultivars.

2.      In Figure 1, The author used 306 SrMV isolates for phylogenetic tree construction, but in the previous text, the author used 932 leaf samples. How to explain?

3.            In lines 103-105, The author claims that the group C was present in four provinces including Fujian (FJ), Guangxi (GX), Hainan (Hainan), and Sichuan, but the group F only occurred in Yunnan (YN) province. What are the unique features of the CP fragments appearing in group F compared to other groups? They should be explained in the article.

4.            The author should provide the total sample size for each province.

5.      The manuscript was not well written with many typos, grammar errors, redundancies, and inaccurate statements. A major revision in both science and language is needed to improve readability and clarity.

Comments on the Quality of English Language

1.      The manuscript was not well written with many typos, grammar errors, redundancies, and inaccurate statements. A major revision in both science and language is needed to improve readability and clarity.

Author Response

Comment 1: In the manuscript of “Phylogeny and Genetic Divergence among Sorghum Mosaic Virus Isolates Infecting Sugarcane”, Which enriches the information of genetic evolution of SrMV world-wide and provides a reference for epidemiology and control of mosaic in sugarcane. These findings enrich the information of genetic diversity of this virus. in my opinion, just a few changes are required. Reply: Thank you for your positive comments. Comment 2: In lines 77-78, The author claims that SrMV is one of main viruses infecting sugarcane particularly modern commercial cultivars, followed by SCSMV in China. Are all the 901 materials used by the author modern commercial cultivars. Reply: Yes, most of leaf samples were collected from modern commercial cultivars. Namely, 839 samples were from modern cultivars (Saccharum spp. hybrids, Sh) and 62 samples were from chewing cane (S. officinarum, So). This information was described in the M&M section. Comment 3: In Figure 1, The author used 306 SrMV isolates for phylogenetic tree construction, but in the previous text, the author used 932 leaf samples. How to explain? Reply: A total of 901 leaf samples with mosaic were collected in this study. The SrMV was detected by RT-PCR in 632 of 901 (70.1%) leaf samples. Subsequently, we just cloned and sequenced the target fragments from partial SrMV-positive PCR products, which generated 265 SrMV CP sequences. A total of 306 CP sequences (this study = 265 and GenBank database = 41) were used for phylogenetic tree construction. Comment 4: In lines 103-105, The author claims that the group C was present in four provinces including Fujian (FJ), Guangxi (GX), Hainan (Hainan), and Sichuan, but the group F only occurred in Yunnan (YN) province. What are the unique features of the CP fragments appearing in group F compared to other groups? They should be explained in the article. Reply: We added the supplement Figure S1 for uncovering the features of the CP fragments among SrMV phylogroups. It is noteworthy that no deletion, but a unique site mutation is present among these CP amino acid sequences in the group SrMV-F compared to other groups. We added these results in the text. Comment 5: The author should provide the total sample size for each province. Reply: We added this information in the legend of Figure 2 and the M&M section. Comment 6: The manuscript was not well written with many typos, grammar errors, redundancies, and inaccurate statements. A major revision in both science and language is needed to improve readability and clarity. Reply: We have seriously improved the English in the whole manuscript under the English-speaking expert.